# The Persistence of Antibiotic Resistance in Observational Studies: Is It Really Due to Differences in Sub-Populations Rather than Antibiotic Use?

**DOI:** 10.3390/antibiotics14010039

**Published:** 2025-01-06

**Authors:** Peter Collignon, John J. Beggs

**Affiliations:** 1Canberra Hospital, Garran, ACT 0200, Australia; 2Medical School, Australian National University, Canberra, ACT 2601, Australia; 3Independent Researcher, Melbourne, VIC 3000, Australia; beggsjo_home@yahoo.com

**Keywords:** antibiotic resistance, antibiotics, persistence, heterogeneity, model, confounders

## Abstract

**Background**: The carriage of resistant bacteria and prior antimicrobial treatment are related, but in an individual, this diminishes over time. To better manage antimicrobial resistance risks, it is crucial that we better untangle any lasting impact of antibiotic use compared to other factors. This understanding is essential for informing antimicrobial stewardship programs and to better manage other important factors that likely contribute to persistently higher rates of antimicrobial resistance in different populations. The true association between antibiotic use and resistance is likely to be significantly overestimated due to the confounding influence of varying infection risk patterns within populations. Though missing explanatory covariates are a well-known cause of falsely interpreted statistical findings, how the problem manifests in this context has a particular and interpretable structure. This issue does not appear to have been previously addressed with clarity. To be more easily understood, a simple model is used to demonstrate this. **Results**: In our theoretical model case study, when we exclude an effect of past antibiotic usage, clinical history alone can predict future resistance patterns. Heterogeneity in infection risk and antibiotic resistance carriage rates, along with consequently observed antimicrobial treatment, often suffice to predict a pattern of resistance that mimics what is assumed to be caused by genuine biologically driven resistance by the associated use of antibiotics. The biological impact and/or lasting effects of antibiotics are not necessary for this prediction. **Conclusions**: Antimicrobial stewardship policies and future research must directly address how much of the apparent persistence of resistant bacteria results from biological consequences of antibiotic use compared to pure statistical confounding arising due to heterogeneous risks in community infection patterns.

## 1. Introduction

Patients previously treated with antibiotics are more likely to carry and be infected by antibiotic-resistant bacteria [1,2,3,4]. This resistance arises from two key biological phenomena: the alteration of the body’s microbiome by antibiotics, promoting resistant bacteria growth, and the survival and increased numbers of “persister” phenotypes through initial treatments [2,3,5]. The link between antibiotic use and resistance is intricate, with empirical evidence indicating that this correlation can persist for years [6], though it typically diminishes much sooner and often has resolved 6 months after antibiotic use [4]. But antibiotic-resistant bacteria are acquired and can persist for long periods of time in travelers, even when they have not taken any antibiotics [7,8]. Thus, there are factors other than just antibiotic use associated with “persistence”.

Global studies reveal that poor infection control, driven by factors like inadequate socio-economic infrastructure, is more strongly associated with antimicrobial resistance (AMR) and its spread than antimicrobial usage itself [9,10]. Factors such as poor governance, corruption, inadequate refrigeration, compromised water quality, and insufficient sanitation contribute to the proliferation of resistant bacteria. Environmental and socio-economic factors create differences in individual infectiousness and risks of becoming infected, which can complicate the understanding of the “persistence” of antibiotic resistance.

This paper aims to explain how a heterogeneous risk of infection among individuals (a familiar complication in several other medical contexts [5,11,12,13,14]) can lead to an observed correlation between resistance and historical antimicrobial use, even when that correlation is not causally related. Hence, it is prudent to take extra care when considering the persistence of antibiotic resistance that is measured from observational studies.

Segregating observed data into sub-samples of similar individuals is one way to reduce heterogeneity in the unobserved characteristics of the sample data. For example, settings with higher infection rates (and acquisition and carriage of resistant bacteria), such as hospitals and aged care facilities, are typically analyzed separately from community data. When creating separate sub-samples is impractical, statistical models aim to incorporate covariates to account for underlying unobserved risk rates.

However, the diversity of microenvironments in any population can be very large. Influences such as work conditions, neighborhood characteristics, housing quality, socio-economic status, ethnicities, schools, and life histories can result in significant individual variations in infection risk and the likelihood of contracting resistant infections. Often, it is challenging to statistically characterize these diverse microenvironments. Even with disaggregated data, identifying sufficient high-quality covariates to mitigate infection risk heterogeneity’s distorting effects remains a formidable task, and it remains a confounding factor when interpreting statistical results from observational studies.

To demonstrate how heterogeneous infection and carriage risks by themselves can mimic aspects of the persistence phenomenon usually attributed to antibiotic use, we employ a simple counterfactual model devoid of biological persistence caused by antibiotic use. Antibiotics obviously do influence rates of antibiotic resistance and persistence. But to see what other factors might also have major effects, our model assumes that prior antibiotic use itself has strictly no effect on the development or persistence of resistant bacteria. This assumption means that prior antibiotic use is assumed to have no effect on either the future risks for any one individual or on general disease carriage rates in the population. This model assumption is adopted to show what other factors can also result in the “persistence” of resistant bacteria in a population. Here, in each sub-population, the circulating carriage rate or prevalence of resistant bacteria is due solely to local socio-economic and environmental factors.

## 2. Results

Here, we distinguish four scenarios that describe what might be known about the medical treatment history and community life of people presenting for treatment. These scenarios are shown in Table 1 and Table 2, where the infection risk profile of an individual in each community over the course of some arbitrary period of time (e.g., six months or one year) is assumed. Table 2 then aligns with what is presented in Table 1.

In clinical practice, what is known about a patient will usually be an amalgam of knowledge from different information sources; however, the simple categorization of scenarios here is useful in clarifying the key insights of this paper. In clinical practice, a common scenario is likely to be one has some knowledge about the patient’s clinical history and where the patient lives (e.g., house, apartment, community nursing home), although usually their total community life experience (interaction with others, etc.) will still not be known.

Scenario 1: The individual’s community identity is known, but their history of prior infection is not known. This is the most straightforward scenario, as the risk of infection remains constant in each period and solely depends on the community to which the individual belongs to. For instance, in our numerical example, if the individual is from Community A, the risk of any infection in each period is consistently (20% + 25%) or 55%, and the risk of resistant infection are 25%. For an individual from Community B, these risks for any infection is (10% + 5%) or 15% and for a resistant infection 5%, (as per Table 1).

This scenario corresponds to knowing the patient is from a nursing home, but the patient comes with no medical history records

Scenario 2: The individual’s community identity is known, and their history of prior infection is known. The risk profiles of this scenario are identical to those in Scenario 1. Even if the clinician is aware of the individual’s prior infection history, it provides no additional information in our model because, by assumption, we are modelling a case where there are no lasting biological effects of previous infection and treatment. Hence, there is no additional useful information contained in the clinical history of infection.

This corresponds to knowing a patient is from a nursing home and the patient comes with their medical history records, including antibiotic and other drug therapy.

Scenario 3: Neither the individual’s community identity nor their history of prior infection is known. This scenario is slightly more complex than Scenario 1 in that health outcome probabilities can be calculated as the population-weighted average of the risks in each community. For instance, the probability of any infection is [0.3 × (0.20 + 0.25) +0.7 × (0.10 + 0.05)] or 24%, and the probability of a resistant infection is [(0.30 × 0.25) + (0.7 × 0.05)] or 11%.

This equates to an unconscious patient presenting to the emergency department with no known history or known place of residence.

Scenario 4: The individual’s community is unknown, but his or her clinical history of infection is known. Importantly, this scenario is analogous to clinical situations where the treating physician is unaware of the patient’s infection risks (here, it is because the community membership is not known) but does have access to the patient’s medical case history. In terms of conventional statistical analysis, this corresponds to missing environmental covariates that identify the individual’s community, factors such as occupational exposure to infection risk, residential circumstances such as poor housing, details of travel to high-risk destinations, etc. Clinical history now becomes relevant because it implicitly contains information on a person’s community and, hence, the underlying risk of resistant infection.

Scenario 4 corresponds to a patient whose entire medical history is available, but a sufficiently detailed biographical history of jobs, residences, travel, etc., is not available.

The analysis that follows uses Scenario 4, the most clinically realistic of the four scenarios to use in our model, to look at the effects if antibiotic use is excluded as having a persistent effect. Scenario 4 corresponds to having patients whose entire medical histories are available on file, but sufficiently detailed demographic and biographical histories of jobs, residences, travel, lifestyle, etc., are not collected either because of the cost of garnering and recording such detailed information or because of patients’ unwillingness to supply such information. Even if much of this information was informally known by the clinician as a result of previous patient contact, such detailed information will rarely be, if ever, recorded in a usable form for ex-post empirical statistical analysis by researchers studying the persistence issue.

The more recently a person was infected, the greater the chance that that person came from the high-risk community. Thus, even when there is no biological dependence of future health outcomes upon prior outcomes, clinical history helps predict future outcomes, and that predictability declines with time since the last infection and treatment. The upcoming numerical example is used to provide a concrete illustration of this phenomenon.

Over four time periods, with three possible outcomes in each period (Not Infected or Infected–Sensitive or Infected–Resistant), there are (3 × 3 × 3 × 3) = 81 permutations of the possible sequences of health outcomes. The discussion that follows focuses on sequences having Infected–Resistant outcomes in the fourth period when outcomes in the prior three periods are known.

In the initial three periods, there are (3 × 3 × 3) = 27 permutations of the possible sequences of health outcomes that can be observed. In each community, the probability of each sequence is calculated by multiplying together the risk of each outcome in the sequence. As an example of the calculation, the historical sequence (Not Infected, Infected–Resistant, Infected–Sensitive) occurs in Community A with a probability =0.55×0.25×0.2=2.75% and in Community B with a probability =0.85×0.05×0.10=4.25%.

In each community, the probability of any three-period sequence followed by Infected–Resistant in the fourth period is simply the probability of that prior sequence multiplied by the probability of Infected–Resistant. Continuing the example above, the probability of the sequence (Not Infected, Infected–Resistant, Infected–Sensitive) followed by Infected–Resistant in the fourth period in Community A is simply = 0.0275 × 0.25 = 0.687%, and in Community B, it is 0.0425 × 0.05 = 0.2125%.

Insights can be obtained by collecting sequences into meaningful groups. For instance, one grouping could be of all sequences of prior history where an individual experiences one each of Not Infected, Infected–Sensitive, and Infected–Resistant in the first three periods. There are six such sequences that make up the groups: (a) Not Infected, Infected–Sensitive, Infected–Resistant; (b) Not Infected, Infected–Resistant, Infected–Sensitive; (c) Not Infected, Infected–Resistant, Infected–Sensitive; (d) Infected–Resistant, Not Infected, Infected–Resistant; (e) Infected–Sensitive, Not Infected, Infected–Resistant; (f) Not Infected, Infected–Sensitive, infected-Resistant).

Consider groupings of sequences that could occur in periods one to three where the following could occur:Sequences are grouped by the number of times an individual has been Infected–Sensitive or Infected–Resistant in the first three periods;Sequences are grouped by the number of periods elapsed since the individual’s most recent infection.

The conditional probability of health outcomes in the fourth period for the above grouping of sequences is presented in Table 3 and Table 4. An explanation of the calculation follows.

The conditional probability of being Infected–Resistant in the fourth period, given one of the defined groups of prior clinical history, can be calculated by a familiar conditional probability formula (this equation is stated more formally in the Appendix A).



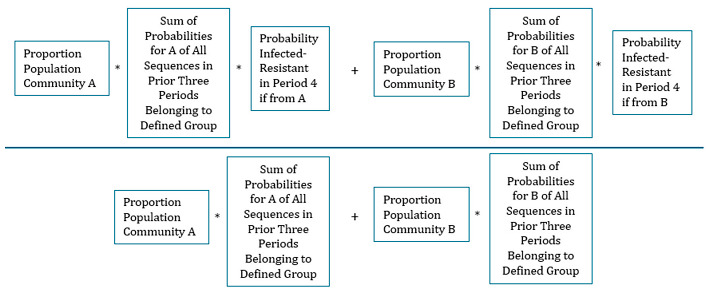



Illustrating the calculation with the example given above, there are six possible prior sequences (permutations) of prior history where a person can be Not Infected in only one period, Infected–Sensitive in only one period, and Infected–Resistant in only one period,
$$\begin{array}{c} Prob(InfectedResistant\;Period\;4\;Given\;Each\;once\;in\;prior\;three\;periods\;Not\;infected,\;Infected\;Sensitive\\ \;\;and\;Infected-Resistant)\;\\ \;=\frac{\;0.3\times \left\{6\times 0.55\times 0.25\times 0.2\right\}\times 0.25+0.7\times \left\{6\times 0.85\times 0.05\times 0.10\right\}\times 0.05}{0.3\times \left\{6\times 0.55\times 0.25\times 0.2\right\}+0.7\times \left\{6\times 0.85\times 0.05\times 0.10\right\}}\;\\ \;=19.7\% \end{array}$$

Table 3 and Table 4 show the conditional probability of being Infected–Resistant in period four, given different descriptions of a person’s clinical history in the preceding three periods.

Table 3 reveals that the probability of being infected by a resistant strain (Infected–Resistant) in period 4 is influenced by both the total number of past infections and the count of those infections that were resistant. This likelihood escalates with an increase in both these factors. This pattern emerges even though the underlying risk to any individual in either community is completely unrelated to their history. In this instance, what mimics the enduring biological effects of previous treatment is merely the fact that a clinical history of more infection means the individual is more likely to belong to the community with a higher infection risk and, if infected, also has an increased risk of resistant infection.

Table 4 illustrates that the likelihood of observing a resistant infection decreases as the time since the last infection increases. This trend mimics the pattern observed in biologically induced antimicrobial resistance, where the effect gradually diminishes as time passes since the individual’s last treatment. However, what appears to be a decline in persistence is merely a result of the fact that individuals with a longer interval since their last infection (and, consequently, their last antimicrobial treatment) are more likely to originate from a community with a low risk of infection and resistance.

## 3. Discussion

Antibiotic resistance is often associated with and therefore linked to antibiotic use [1,2,3,4,5,6]. But this correlation may not need to be all due to the direct effects of the antibiotics themselves. Our model shows that this same pattern can also be created by the conflating effects of unmodelled heterogeneity in the underlying risk factors. This paper shows how varying infection risks among individuals could explain the observed persistence of resistance and where it is not due to antibiotic use. Additionally, for purely statistical reasons, the link between antibiotic use and resistance tends to weaken with time, affecting the underlying subpopulations.

Well-designed statistical studies can reduce the impact of community heterogeneity on observed persistence by including variables that account for individual risks. Identifying variables that fully capture the diverse infection risks across communities is challenging, and as a result, there may always be a tendency for past resistance to predict future resistance, independent of the biological effects of antibiotics.

Importantly, if the goal is only to inform clinical decisions before testing, the source of statistical persistence is less important. A model that predicts the degree of persistence is valuable, even without a biological link between antibiotic use and resistance.

However, when formulating antibiotic stewardship programs for empirical antibiotic use, it is crucial to recognize that the correlation between antibiotic use and resistance need not be due solely to antibiotic-induced changes in the microbiome. The heterogeneous spread and acquisition of resistant bacteria, or community “contagion”, is another significant factor that is easily overlooked. Unless research can confidently differentiate between (a) biological causes of persistence and (b) conflating statistical consequences of heterogeneous infection risk in sub-populations, it is prudent when considering observational studies to be cautious when interpreting the role of past antibiotic use as the major factor in explaining observed rates of antibiotic resistance. By not considering other factors besides antimicrobial use, observational studies can often too quickly jump to a causal conclusion [15,16,17,18,19,20,21,22,23,24]. A large study from Israel looked at factors other than antibiotic use to predict antibiotic resistance in bacteria causing urinary tract infections. It showed that the persistence of AMR and predictability of resistance in the future for individuals, while associated with prior antibiotic use, was also affected by many other factors involved, including age, gender, and place of residence [1].

This may also explain why, in many large population studies, the volumes of antibiotics used, or variations in their usage volumes, have had either no effect or only minor effects on the levels of associated antimicrobial resistance rates seen [21,22,23,24]. Two studies looked at the impact of changes in antibiotic resistance rates or trends in *E. coli* associated with decreases in antibiotic usage of a population [23,24], but no falls were seen in antibiotic resistance after reductions in antibiotic usage. In recent global studies, there was a poor correlation between antibiotic consumption and AMR levels [9,25]. In developed regions, such as Europe, a correlation with antibiotic usage was seen, but this was not evident when using global data [9]. The underlying socio-economic factors in countries or populations seem to have much higher impacts on resistance rates in comparison to antibiotic usage volumes [9,10], as do factors such as regions where people live [16,20,21,22].

This paper addresses one important aspect of the complex set of factors involved in understanding the persistence of antibiotic resistance. A simplified model was used for clarity, but that abstraction was not meant in any way to deny the biological consequences of antibiotic use. An effect of antibiotic use was left out of our model to show other factors that are involved in both rates of resistance seen and in the persistence of resistant bacteria. It is certain that more realistic subsequent theoretical models, along with further empirical research, will shed more insights into the significance of the issue raised here.

## 4. Methods

Observational data are collected when individuals who are infected seek medical treatment and undergo testing. Imagine a population made up of two communities, A and B. In each period, individuals in Community A have a higher likelihood of becoming infected and, if infected, are more likely to have a resistant infection. Importantly, we assume for the sake of this model that there is no biological link between prior antibiotic use and the risk of antimicrobial resistance, either at the individual or community level. Additionally, the risk of infection for any individual is assumed to be constant in each time period.

To make the discussion more tangible, we construct a numerical example. Suppose the two communities, A and B, make up 30% and 70% of the total population, respectively. Additionally, assume that the infection risk profile of an individual in each community over the course of some arbitrary period of time (e.g., six months or one year) aligns with what is presented in Table 1. Observational data are typically generated when those who are infected present themselves clinically for treatment. The insights generated in this paper arise from the implications of elementary probability calculations discussed in the Section 2.

## 5. Conclusions

The statistically observed persistence of antimicrobial resistance in observational studies is affected by factors other than just antimicrobial use. Antimicrobial stewardship policies and future research must directly address how much of the apparent persistence of resistant bacteria results from biological consequences of antibiotic use compared to pure statistical confounding arising due to heterogeneous risks in community infection patterns.

## Figures and Tables

**Table 1 antibiotics-14-00039-t001:** Assumed prevalence of health outcomes in each period in each community.

Health Outcome	Probability Community A	Probability Community B
Not Infected	55.0%	85.0%
Infected with Sensitive Bacteria	20.0%	10.0%
Infected with Resistant Bacteria	25.0%	5.0%

**Table 2 antibiotics-14-00039-t002:** Information known about the patient: four scenarios.

	Prior Treatment History **Not Known**	Prior Treatment History **Known**
Identity of community (home, work, recreation etc) **Known**	Scenario 1	Scenario 2
Identity of community (home, work, recreation etc) **Not Known**	Scenario 3	Scenario 4

**Table 3 antibiotics-14-00039-t003:** Scenario 4: Probability of being Infected–Resistant in period 4, given history in the prior three periods.

	Number of Times Having Resistant Infection in Prior Three Periods			
Number of times infected (Sensitive or Resistant) in in prior three periods	None	Once	Twice	Three times
None	7.1%			
Once	10.3%	14.5%		
Twice	15.5%	19.7%	22.5%	
Three Times	20.5%	22.9%	24.1%	24.6%

**Table 4 antibiotics-14-00039-t004:** Scenario 4: Probability of being Infected–Resistant in period 4, given the number of periods since previous infection.

Periods Since Most Recently Infected (Either Infected Sensitive or Infected Resistant	Probability Infected-Resistant in Period 4
Never Infected	9.1%
Three	12.7%
Two	14.3%
One	15.9%

## Data Availability

Data are contained within the article and Appendix A.

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
