# Peer review of "The Persistence of Antibiotic Resistance in Observational Studies: Is It Really Due to Differences in Sub-Populations Rather than Antibiotic Use?"

_antibiotics, 2025, doi:10.3390/antibiotics14010039_

Round 1
Reviewer 1 Report
Comments and Suggestions for Authors
1. Title; the title of the paper can be improved upon to reflect the type of study, for example, ''Persistence of Antibiotic Resistance in Observational Studies: A Theoretical Model Case Study''
2. The model used should be clearly stated in the abstract
3. Line 261-263 needs referencing
4. Conclusion should be used in the abstract not interpretation
5. Conclusion in the text should be more concise. The authors were stating limitations of the model used rather stating the clear finding from the outcome of the model
Author Response
- Title; the title of the paper can be improved upon to reflect the type of study, for example, ''Persistence of Antibiotic Resistance in Observational Studies: A Theoretical Model Case Study''
We think current title is better as it better present our findings; but happy to add “A Theoretical Model Case Study” as a subtitle
- The model used should be clearly stated in the abstract
We have added the words “Theoretical Model Case Study” in the abstract as suggested by the reviewer.
- Line 261-263 needs referencing
These have now been added.
- Conclusion should be used in the abstract not interpretation
Changed as suggested by the reviewer
- Conclusion in the text should be more concise. The authors were stating limitations of the model used rather stating the clear finding from the outcome of the model
We agree. We have now added a new conclusion at the end of the discussion section. We have left the previous text of the conclusion in the last part of the discussion as we need to show the limitations in our study. Antibiotic use does cause resistant bacteria to develop, and these can persist. But we think also as we show via our study, and as the reviewer points out, that observational studies can too quickly jump to a causal conclusion. We have added a comment to our discussion.
Reviewer 2 Report
Comments and Suggestions for Authors
1- In the study, care should be taken to give the full name of the abbreviations where they are first used. 2- Tables and their names should be adapted to the writing style of a scientific article. 3- The authors used very few references in the introduction and other sections. References related to this study should be added. 4- Many equations were used in the study, but these equations were not numbered. 5- The difference of the study from similar studies and its contribution to the literature should be explained more clearly. 6- Can the proposed model be used for all antibiotics?
Author Response
- 1. In the study, care should be taken to give the full name of the abbreviations where they are first used.
- We fully agree with this view. But as far as we are aware the only abbreviation, we have used in out paper was AMR in the abstract and then in the discussion. This has now been replaced by the word antimicrobial resistance when first used in either section.
- 2. Tables and their names should be adapted to the writing style of a scientific article.
- We think they are already but will ask the editorial staff to help us put the titles in the style this journal thinks best.
- We think they are already but will ask the editorial staff to help us put the titles in the style this journal thinks best.
- 3. The authors used very few references in the introduction and other sections. References related to this study should be added.
- We have added many extra references and as suggested by reviewer1 as well.
- 4. Many equations were used in the study, but these equations were not numbered.
- We are not sure how to best handle this. In the paper itself there are calculations rather equations, and mainly under the heading of the different scenarios. So to some degree they are already at least grouped by being kept under specific headings. We think numbering each calculation will make the paper more difficult to read, rather than easier. But we will differ to the editorial staff’s view and advice on this.
- 5. The difference of the study from similar studies and its contribution to the literature should be explained more clearly.
We are unaware of any previous studies similar to our addressing this issue. Our current conclusion we believe outline the contribution to the literature as well what is in the abstract.
In essence, antibiotic use does cause resistant bacteria to develop, and these can persist. but we show in our study, is the that observational studies where different population have different prevalence of AMR present, can too quickly many likely jump to a causal conclusion that all AMR persistence is due to antibiotic use.
- 6- Can the proposed model be used for all antibiotics?
We think it can, and also for most bug/drug combinations as implied in our conclusion and suggestions for further research. The difficulty will be as we outline, having sufficient individual patient data, including their risk stratification for where they live and traveling etc to be in or reside in high AMR prevalence situations plus their antibiotic usage.
Reviewer 3 Report
Comments and Suggestions for Authors
The authors have presented an important idea very clearly with an abstract example. The manuscript is very well written and the conclusion well argued. I have a concern regarding the broader acceptance of the piece. It may help the reader to further emphasise that AMR can arise through antibiotic overuse and misuse but that observational studies can too quickly jump to a causal conclusion. Perhaps some reference to case studies or laboratory evidence might keep the reader from misinterpreting the conclusions. That is state the good evidence for the link before giving this well worked example of how observational studies are not so straightforward to interpret.
Author Response
The authors have presented an important idea very clearly with an abstract example.
Thank you
The manuscript is very well written and the conclusion well argued.
Thank you
I have a concern regarding the broader acceptance of the piece. It may help the reader to further emphasise that AMR can arise through antibiotic overuse and misuse but that observational studies can too quickly jump to a causal conclusion.
We agree and have tried to better highlight this in our new conclusion at the end of our discussion and on line 378 we have in effect added the reviewer’s succinct comments.
Perhaps some reference to case studies or laboratory evidence might keep the reader from misinterpreting the conclusions.
Unfortunately there are almost no papers on any large population to refer to. Hence why we addressed the issue in this paper. There is a study from Isreal (our current reference 1), that showed that persistence of AMR and predictability of resistance in the future for individuals while associated with prior antibiotic use, they also showed that there were many other factors involved including gender and place of residence. We have added more discussion about that paper to better make this point and referenced other added papers that show place of residence is important.
That is state the good evidence for the link before giving this well worked example of how observational studies are not so straightforward to interpret
Thank you. We agree and why we thought writing this paper was important.